# High-Cycle Fatigue Behaviour of the Aluminium Alloy 5083-H111

**DOI:** 10.3390/ma16072674

**Published:** 2023-03-28

**Authors:** Branko Nečemer, Franc Zupanič, Tomaž Vuherer, Srečko Glodež

**Affiliations:** Faculty of Mechanical Engineering, University of Maribor, Smetanova 17, 2000 Maribor, Slovenia

**Keywords:** aluminium alloys, rolling direction, high-cycle fatigue, fracture analysis

## Abstract

This study presents a comprehensive experimental investigation of the high-cycle fatigue (HCF) behaviour of the ductile aluminium alloy AA 5083-H111. The analysed specimens were fabricated in the rolling direction (RD) and transverse direction (TD). The HCF tests were performed in a load control (load ratio *R* = 0.1) at different loading levels under the loading frequency of 66 Hz up to the final failure of the specimen. The experimental results have shown that the S–N curves of the analysed Al-alloy consist of two linear curves with different slopes. Furthermore, RD-specimens demonstrated longer fatigue life if compared to TD-specimens. This difference was about 25% at the amplitude stress 65 MPa, where the average fatigue lives 276,551 cycles for RD-specimens, and 206,727 cycles for TD-specimens were obtained. Similar behaviour was also found for the lower amplitude stresses and fatigue lives between 10^6^ and 10^8^ cycles. The difference can be caused by large Al_6_(Mn,Fe) particles which are elongated in the rolling direction and cause higher stress concentrations in the case of TD-specimens. The micrography of the fractured surfaces has shown that the fracture characteristics were typical for the ductile materials and were similar for both specimen orientations.

## 1. Introduction

Aluminium alloy series 5xxx is a group of alloys based on aluminium–magnesium (Al-Mg) compositions [1]. These alloys are known for their high strength-to-weight ratio, excellent corrosion resistance, and good formability [2,3,4]. Based on these characteristics and their mechanical properties, these alloys are used widely in the automotive and aerospace industries, transportation, and other applications where lightweight and corrosion-resistant materials are required [5,6,7,8,9].

In the past, numerous studies have been conducted to understand the fatigue behaviour of AA 5xxx alloys [10,11,12,13,14]. Fatigue is an essential factor in determining the behaviour of mechanical parts under variable loads and is the primary cause of 80–90% of engineering failures. In applications that use aluminium alloys frequently, it is necessary to understand the fatigue performance of the components and the effects of operating parameters on the fatigue behaviour of the analysed constructional components. A fatigue assessment can be carried out using the stress–life (S–N) or the strain–life approach, depending on whether stresses in the critical cross-sections of the analysed component are in the elastic or plastic areas [15,16]. In the case that only elastic stresses occur, the S–N approach is usually used to obtain the fatigue life up to the final failure of the component. Establishing comprehensive databases, including S–N information, is essential for evaluating the fatigue characteristics of components under different operating conditions accurately. The fatigue life of Al-alloys is influenced not only by material characteristics but also by the characteristics of the specimens, such as microcavities formed during production, surface defects, hot or cold deformation, and changes in grain structure [17,18,19]. The tensile strength and fatigue life of Al-alloys are affected slightly by the rolling direction at room temperature, but the effects become more significant at higher temperatures due to the expansion of the grain structure [20,21]. Hockauf et al. [22] investigated the effect of different precipitate morphologies on low-cycle fatigue (LCF) and fatigue crack growth in an ultrafine-grained (UFG) age-hardening aluminium alloy. Their experimental results showed that the newly formed, coherent precipitates in the thermally recovered condition contribute to more planar slip behaviour, slip localisation and early failure during LCF loading. In [23], the researchers investigated the high-cycle fatigue behaviour of an Al-Si-Cu-Mn aluminium alloy at room temperature and 350 °C and evaluated the effect of different second phases on its fatigue behaviour. The alloy demonstrated excellent fatigue resistance at both temperatures, with a high-cycle fatigue strength of 125.0 MPa at room temperature and 47.5 MPa at 350 °C. The study found that fatigue cracks in the alloy at room temperature originated from casting defects, while at 350 °C, they nucleated from the primary Al-Mn-Si phase on the specimen’s surface. The study also demonstrated that the modulus difference between the Al-Mn-Si phase and the alpha aluminium phase was higher at 350 °C than at room temperature, leading to more crack initiation and propagation along the phase interface at 350 °C. Khisheh et al. [24] investigated the influences of surface roughness and heat treatment on the high-cycle bending fatigue properties of A380 aluminium alloy under stress-controlled cyclic loading. The results showed that the heat treatment increased the high-cycle bending fatigue lifetime by 26% (for the highest stress level) and 85% (for the lowest stress level), respectively. The study also found that samples with low roughness had a longer fatigue life than those with high roughness. In the research presented by Gao et al. [25], the authors investigated the effect of surface mechanical attrition treatment (SMAT) on the fatigue performance of a 7075-T6 aluminium alloy in both high-cycle fatigue (HCF) and very-high-cycle fatigue (VHCF) regimes. The results showed that SMAT could improve fatigue strength in the HCF regime but decrease it in the VHCF regime. Meng et al. [26] investigated the vibration fatigue improvement of 2024-T351 aluminium alloy by ultrasonic-assisted laser shock peening (ULP). The results showed that the ULP increases the dislocation density significantly and refines the grains, leading to increased compressive residual stress and microhardness and inhibited crack initiation and propagation, resulting in a significant increase in vibration fatigue life. Xing et al. [27] studied the transition fatigue failure of weld toe cracking and weld root cracking in aluminium fillet welds. Their experimental results showed that the weld root cracking has a lower fatigue life and wider scatter band than the weld toe cracking. The researchers also proposed the weld sizing criterion for avoiding weld root cracking in fillet-welded aluminium connections. Sakin et al. [28] investigated the bending fatigue lives of AA 1100 and AA 1050 aluminium sheets experimentally under both high-cycle fatigue (HCF) and low-cycle fatigue (LCF) conditions. The specimens were tested along four different directions, including the rolling direction (RD) and transverse direction (TD). The results showed that the longest fatigue lives in the LCF region were observed in the AA1050 (RD) specimens, while the AA 1100 (RD) specimens had the longest fatigue lives in the HCF region. In the research works presented in [29,30,31,32], the researchers investigated the fatigue behaviour of cellular structures made of different aluminium sheets. The specimens were fabricated in a rolling direction using water jet cutting technology.

Material testing provides valuable information on the material’s mechanical proper-ties, such as its stress–strain response, deformation, fatigue life, and fracture behaviour. This information is crucial for optimising the design and manufacture of engineering structures and components made from this material and assessing their performance and durability in various applications. The proposed study is the continuation of the author’s previous work [33], where the LCF behaviour of the aluminium alloy 5083-H111 was investigated. The obtained experimental results (cyclic S–N curve, LCF-fatigue parameters) were then used in work [34], where Nečemer et al. analysed the LCF behaviour of auxetic cellular structures. As presented by Lehmus et al. [35], cellular structures represent a unique opportunity for adoption in lightweight design due to their favourable characteristics regarding sound isolation, damping, energy absorption, etc. Using the advanced additive manufacturing (AM) technologies, abrasive water jet (AWJ) cutting technology, etc., different types of cellular structures (see two examples in Figure 1) can be manufactured for specific mechanical properties and other characteristics useful for different engineering applications. When analysing the fatigue behaviour of cellular structures made of aluminium alloy AA 5083-H111, the effect of rolling direction on the fatigue life may be significant, especially in the high-cycle fatigue (HCF) area. For that reason, the proposed research considers the influence of the rolling direction on the fatigue life and fracture behaviour of aluminium alloy AA 5083-H111 in the HCF regime. Additionally, the obtained results could help engineers make the appropriate decisions about the use and performance of this alloy in various engineering applications.

## 2. Materials and Methods

### 2.1. Material and Specimen Geometry

In the experimental investigation, the specimens were made from the aluminium alloy AA 5083-H111, which is a commonly used aluminium alloy known for its excellent corrosion resistance and high strength-to-weight ratio. The chemical composition of this alloy is presented in Table 1.

The flat tensile specimen shown in Figure 2a, taken from one batch of material, was used in the uniaxial quasi-static tensile tests and high-cycle fatigue (HCF) tests. All the specimens were fabricated using the abrasive water jet (AWJ) cutting technology from a 4 mm thick sheet. The specimens were fabricated in the rolling direction (RD) and transverse direction (TD); see Figure 2b.

Figure 3 shows the backscattered electron images of the investigated alloy in its as-received condition for both orientations (in the rolling and transverse directions). The alloy is composed of an Al-rich solid solution, alpha-Al, dark Mg_2_Si particles, large bright iron-rich Al_6_(Fe, Mn) particles, and smaller plate-like Mn-rich Al_6_(Fe, Mn) particles. The particles were identified using EDS. A detailed investigation of particles was carried out in our previous publication [33], and the results were consistent with the findings of Liu et al. [13]. The microstructures in both sections are very similar; however, the dimensions of Fe-rich Al_6_(Fe, Mn) particles are longer in the direction of rolling.

### 2.2. Experimental Methods

In the scope of the experimental investigation, the quasi-static tensile tests were performed, as well as the HCF tests. All the tests were carried out on the electrodynamic pulsating testing machine Vibrophore 100 ZwickRoell.

The quasi-static tensile tests were performed in a force control at an ambient temperature of 22 °C. The force was monitored with a 100 kN tensile–compressive load cell mounted on the testing machine, while the strain was controlled on the narrowed part of the specimen with an axial mechanical extensometer ZwickRoell DigiClip 40. Four quasi-static tensile tests were performed for each specimen orientation.

The HCF tests were performed in load control at different load levels at the same ambient temperature of 22 °C. The loading frequency was fixed to 66 Hz considering the sinusoidal wave shape with the constant load ratio *R* = 0.1. In the experimental testing, eight different values of maximum force *F*_max_ were selected for each specimen orientation. The first stress level (maximum force *F*_max_) was defined at approximately 50% of the yield stress of the material, and each next stress level was reduced by 3 %. At least two or three specimens were tested for each stress level, up to the final failure of the specimen. The run-out condition (without failure) was set to 10^8^ cycles.

The metallographic samples were prepared by grinding with SiC paper and polished using diamond paste. They were examined using the scanning electron microscope (SEM) Quanta 3D (FEI, Eindhoven, the Netherlands). The fractography study was carried out using a light microscope, an Olympus EP 50, and the aforementioned SEM. The microchemical analysis of the particles in a scanning electron microscope Sirion 400 NC (FEI, Eindhoven, The Netherlands) equipped with an energy-dispersive spectrometer (Oxford Analytical, Bicester, UK).

## 3. Results and Discussion

### 3.1. Quasi-Static Tensile Tests

Figure 4 shows the engineering stress–strain curves for both specimens’ layouts. The average mechanical properties of the analysed aluminium alloy AA 5083-H111 are summarised in Table 2. The Young’s modulus for the rolling direction (RD) was evaluated at around 70.8 GPa, while, for the transverse direction (TD), it was around 71.2 GPa. Furthermore, the yield stress and ultimate tensile strength were found to be higher for the RD if compared to the TD. Finally, the strain at fracture was found to be almost the same for both specimens’ orientations.

Based on the experimental results presented in Figure 4 and Table 2, it can be concluded that the mechanical properties (Young’s modulus, yield stress, ultimate tensile strength, elongation at fracture) were quite similar for both specimens. However, the specimens manufactured in the longitudinal/rolling direction demonstrated slightly better properties (except for Young’s modulus, *E*) if compared to the specimens manufactured perpendicular to the rolling direction (transverse direction).

Figure 5 shows the fractured surfaces of the tensile specimens for both directions. The elongation at fracture was about 18 %, while the contraction was very low. The fracture surface was inclined about 45 ° in relation to the loading direction. The fracture shows a ductile character with many dimples. The diameters of the larger pores were 20 to 30 μm, and the smaller ones were a few micrometres. Inside the larger voids, coarse intermetallic particles were present. Inside the smaller pores, some particles were found occasionally. Thus, the pores started to form at the interfaces between the matrix and intermetallic particles. In addition, some of the larger particles also fractured during deformation. The pore sizes were several times larger than the particle sizes.

### 3.2. High-Cycle Fatigue Tests

The high-cycle fatigue (HCF) tests were performed in a force control under pulsating loading (load ratio *R* = 0.1) at a constant frequency of 66 Hz. Eight loading levels were selected, to obtain the S–N curves for both specimens’ layouts. The S–N curves presented in Figure 6 and Figure 7 were defined by plotting the stress amplitude σ*_a_* versus the number of cycles to failure *N_f_*. Some scatter of the experimental results presented in Figure 6 and Figure 7 is evident for both specimens’ layouts. However, the scatter is greater for the transverse direction. Furthermore, the S–N curves are “bilinear” and consist of two linear curves with different slopes, which intersect at the knee point *N_k_* [36]. The fatigue behaviour below and upper the knee point *N_k_* can be expressed as follows:(1)$$\sigma_{a}=\sigma_{a,k}\cdot {\left(\frac{N_{k}}{N_{f}}\right)}^{\frac{1}{k}}\;\mathrm{below}\;\mathrm{the}\;\mathrm{knee}\;\mathrm{point}\;$$
(2)$$\sigma_{a}=\sigma_{a,k}\cdot {\left(\frac{N_{k}}{N_{f}}\right)}^{\frac{1}{k^{*}}}\;\mathrm{above}\;\mathrm{the}\;\mathrm{knee}\;\mathrm{point}\;$$

In Equations (1) and (2), the *k* represents the slope before the knee point, $\sigma_{a,k}$ is the stress amplitude at the knee point, *N_k_* is the number of cycles at the knee point, *k^*^* is the slope after the knee point, and $\sigma_{a}$ is the amplitude stress. The material parameters related to the Equations (1) and (2) are summarised in Table 3.

Figure 8 shows the comparison of S–N curves for both specimen layouts. It is clear that the amplitude stress of the knee point, σ*_a,k_*, is almost the same for both specimen layouts. However, the number of cycles of the knee point is higher for the specimens oriented in the rolling direction, which is then reflected in the slopes *k* and *k^*^*. When analysing the fatigue live for both specimen orientations, it is evident from Figure 6, Figure 7 and Figure 8 that RD-specimens demonstrate longer fatigue live if compared to the TD-specimens. This difference is about 25 % at the amplitude stress 65 MPa, where fatigue lives 276,551 cycles for RD-specimens, and 206,727 cycles for TD-specimens were obtained. Similar behaviour may also be found for the lower amplitude stresses and fatigue lives between 10^6^ and 10^8^ cycles. The difference can be caused by large Al_6_(Mn,Fe) particles which are elongated in the rolling direction and cause higher stress concentrations in the case of TD-specimens.

Figure 9 shows the fractured surfaces of the selected fatigue specimens. The cracks are typically initiated at specimen edges and then propagated in the directions indicated by the arrows. As can be seen, they propagated in the direction parallel to the original sheet surface. In our case, the cracks started at a corner at the top and then propagated towards the bottom.

The initial parts of the cross-sections were relatively flat and approximately normal to the loading direction. At some distances, the fracture surface became rougher, and this coincided with the distortion of the specimen in the bottom part caused by the plastic deformation. The initially rectangular shape became distorted, and the distortion was higher at higher stress levels for the samples on the left side of the images.

Figure 10 and Figure 11 show the scanning electron micrographs of the fracture surfaces of the selected specimens at three different positions. The fracture characteristics were similar irrespective of the sample orientation (longitudinal or transverse layout) and the number of cycles to fracture. At the beginning of the cracks, the surface was rough on the micrometre scale. The crack followed the glide planes in the grains and formed rather flat facets [37]. Striations were present on the facets. The distances between them were less than one micrometre, typically between 0.4 and 0.7 μm (see Figure 12a).

In the last stages of the crack propagation, the surface was rather rough, and it was not perpendicular to the loading direction. The distances between striations increased and were typically between 2 and 4 μm. In these regions, not only striations were present, but also some voids. The voids formed at larger inclusions in the microstructure. They formed in the regions in front of the fatigue crack as the stresses surpassed the yield stress of the material, and they grew until the main fatigue crack overtook them. The final forced residual fracture was almost the same as the tensile fracture (see Figure 3) and consisted of voids of different sizes.

Figure 12 shows some fractured surfaces at a much higher magnification and resolution. The distances between the striations—the striation wavelengths—were measured on the basis of such micrographs. They are presented in Figure 13. The striations were not visible up to distances below 1 mm from the crack initiation site (Figure 12a), which was typically at the specimen corner. This is Region A of crack propagation, which is highly sensitive to microstructure characteristics [38]. Usually, a crack with a length 1–2 mm is considered a small fatigue crack. One can observe grain boundaries and facets in the crystal grains, which typically glide planes. The crack propagation rate is often in the order of nm, and the specimen survives in this region for most of its lifetime [39]. Striations become visible in range B of crack propagation [38]. The striations become stable, and in the first part, their wavelength is in the order of 100 nm; we measured 200 nm at a distance of 1.4 mm from the crack initiation site (Figure 12b).

With the growing fatigue crack, the stress intensity factor grows, and the striation wavelength increases in micrometre size. It should be stressed that with the wavelength of 1 μm, the fatigue crack will proceed 1 mm after 1000 cycles and leads to a quick fracture of the specimen. This wavelength was achieved approximately 3.5 mm from the crack initiation (Figure 12c). In Regime B, the crack propagation is less sensitive to microstructure. During the main part of the crack path, the distances between the striations were larger for the transverse specimen, which can also explain the shorter lifetime for this specimen. The last two measurements were close to the crack ends, where the surface resembled those to that shown in Figure 10c,d and Figure 11c,d, where the scatter was rather large. At the transition from Regime B to Regime C, the crack propagation becomes more microstructure sensitive. In the case of this alloy, fatigue cracks can form even in the front of the main cracks around the largest inclusions, as was discussed before.

## 4. Conclusions

The comprehensive experimental investigation of the high-cycle fatigue (HCF) behaviour of the ductile aluminium alloy AA 5083-H111 was presented in this study. Based on the metallographic investigation of the material, quasi-static tensile tests, high-cycle fatigue tests, and fractography of the fractured surfaces of the quasi-static and fatigue specimens, the following conclusions can be made:The analysed AA 5083-H111 alloy is composed of an Al-rich solid solution, alpha-Al, dark Mg_2_Si particles, large bright iron-rich Al_6_(Fe, Mn) particles, and smaller, plate-like Mn-rich Al_6_(Fe, Mn) particles. The microstructures in both the longitudinal/rolling (RD) and transversal (TD) directions were found to be very similar. However, the dimensions of the Fe-rich Al_6_(Fe, Mn) particles were longer in the longitudinal/rolling direction.The mechanical properties (yield stress, ultimate tensile strength, elongation at fracture) were quite similar for both specimens. However, the specimens manufactured in the longitudinal/rolling direction demonstrated slightly better properties (except for Young’s modulus, *E*) if compared to the specimens manufactured perpendicular to the rolling direction (transverse direction).The experimental results have shown that the S–N curves of the analysed Al-alloy consist of two linear curves with different slopes, which intersect at the knee point *N_k_*. The corresponding amplitude stress at the knee point, σ*_a,k_*, was found to be almost the same for both specimen layouts, while the number of cycles at the knee point, *N_k_*, was found to be higher for the specimens oriented longitudinally to the rolling direction. The difference can be caused by large, in the rolling direction elongated Al_6_(Mn, Fe) particles, which cause higher stress concentrations when tested in the TD. Furthermore, the main part of the larger particles has a cuboidal shape, with a larger axis approximately parallel to the rolling direction. Thus, in the RD orientation, the larger axis of particles lay in the direction of the load and in the TD orientation perpendicular to the load. It could be expected that at TD orientation, higher stress concentrations occurred at the particle–matrix interface and that this leads to slightly worse fatigue resistance in the TD direction.The micrography of the fractured surfaces of the fatigue specimens has shown that the fracture characteristics are similar for both specimen orientations (longitudinal or transversal). The fracture surface has a typical appearance for the ductile material, characterised by striations during propagation of the fatigue crack and final ductile fracture. The distance between striations increased from the crack beginning (less than 0.5 micrometres) to the crack end (more than 3 micrometres).In the proposed research work, we analysed only two specimen orientations: (i) in the rolling direction and (ii) transverse to the rolling direction. The third specimen orientation (45° in regard to the rolling direction) could be investigated in our further research work. Furthermore, further research work should consider the higher number of experiments, especially in the long-life fatigue area (more than 10^7^ loading cycles). In this case, a comprehensive statistical evaluation could be performed to obtain more qualitative results regarding the fatigue behaviour of the analysed aluminium alloy.

## Figures and Tables

**Figure 1 materials-16-02674-f001:**
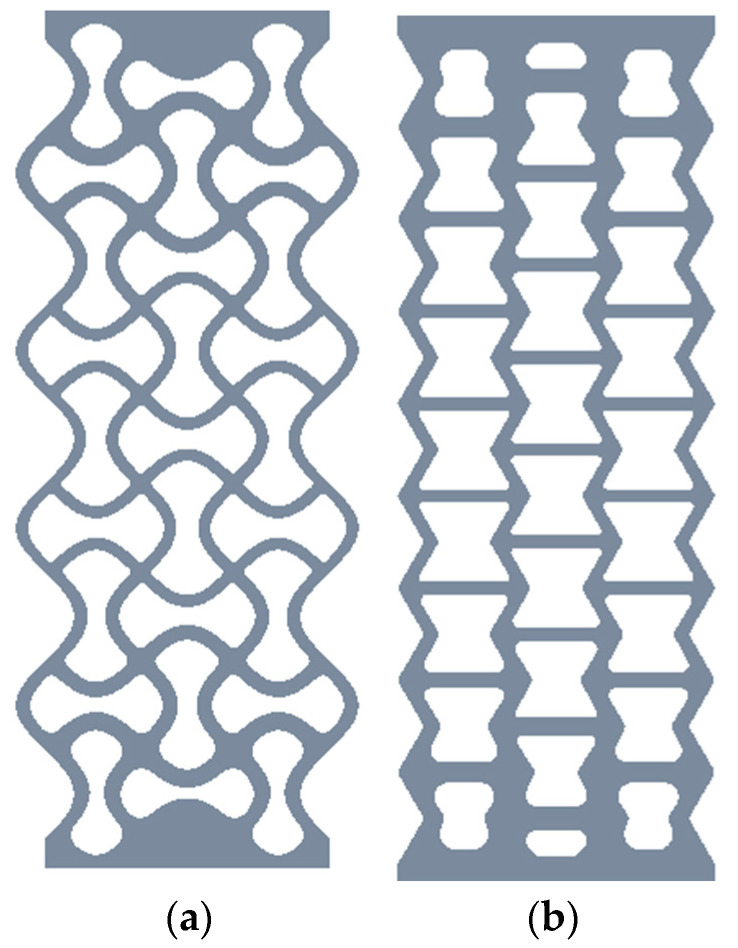
Two examples of cellular structures: (**a**) chiral structure; (**b**) re-entrant structure.

**Figure 2 materials-16-02674-f002:**
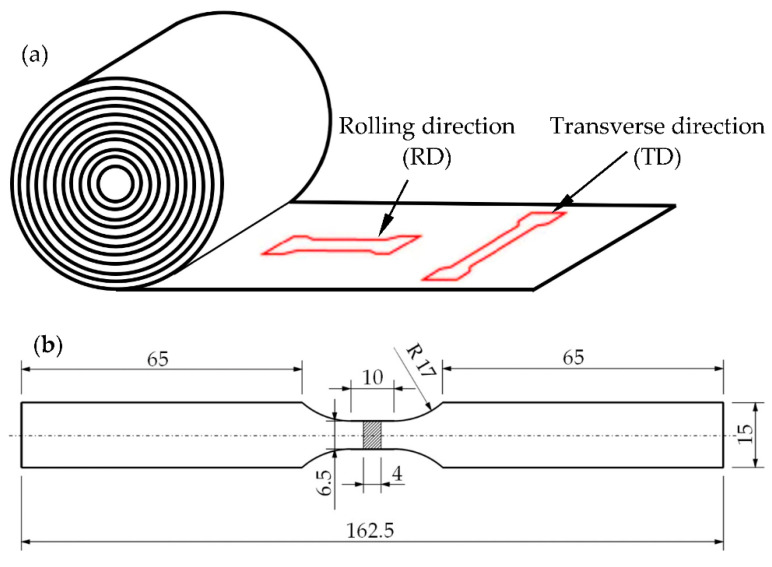
(**a**) Specimen’s orientations. (**b**) Geometry and dimensions of the specimen.

**Figure 3 materials-16-02674-f003:**
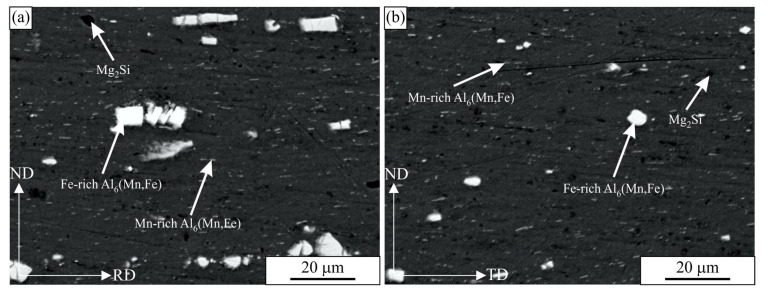
The backscattered electron images of the AA 5083-H111 alloy. (**a**) In the longitudinal direction. (**b**) In the transverse direction: RD-rolling direction, ND-normal direction, TD-transverse direction.

**Figure 4 materials-16-02674-f004:**
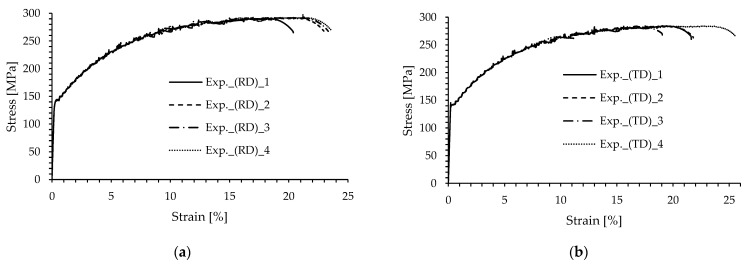
Engineering stress–strain curves of the AA 5083-H111 alloy. (**a**) Rolling direction (RD). (**b**) Transverse direction (TD).

**Figure 5 materials-16-02674-f005:**
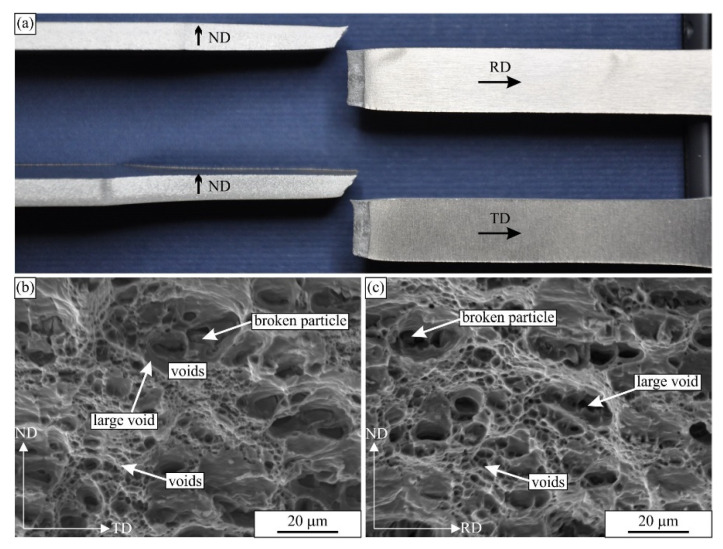
The fractured tensile specimens. (**a**) The photography of the samples. The secondary electron micrographs of the fractured surfaces of the tensile specimens: (**b**) in the RD and (**c**) in the TD.

**Figure 6 materials-16-02674-f006:**
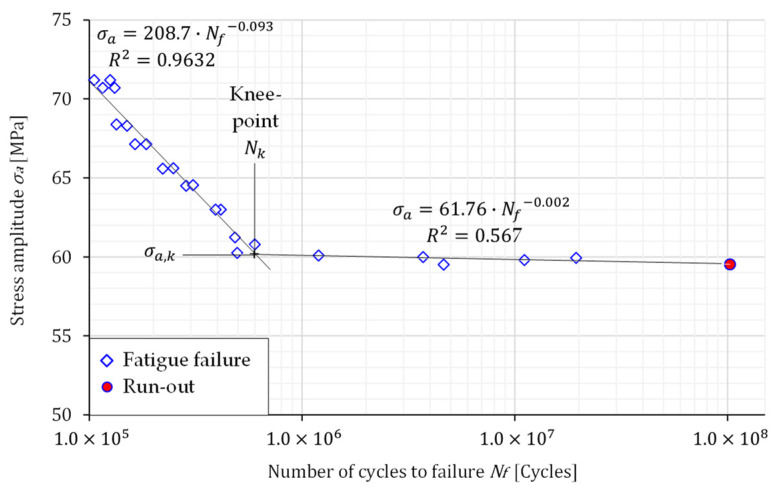
The S–N curve for the rolling direction.

**Figure 7 materials-16-02674-f007:**
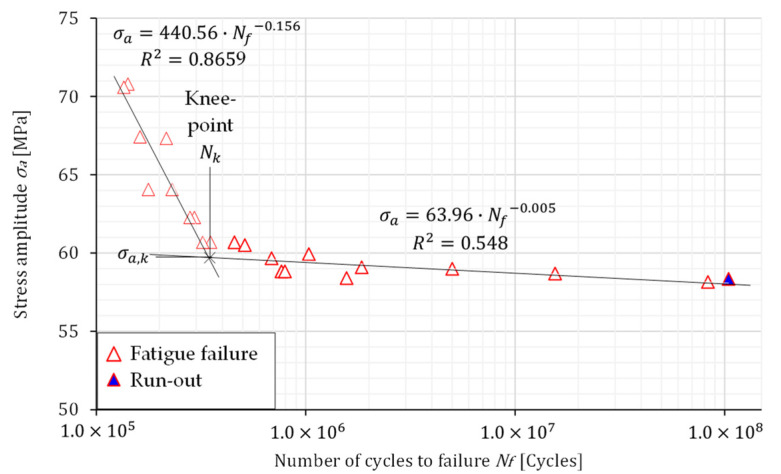
The S–N curve for the transverse direction.

**Figure 8 materials-16-02674-f008:**
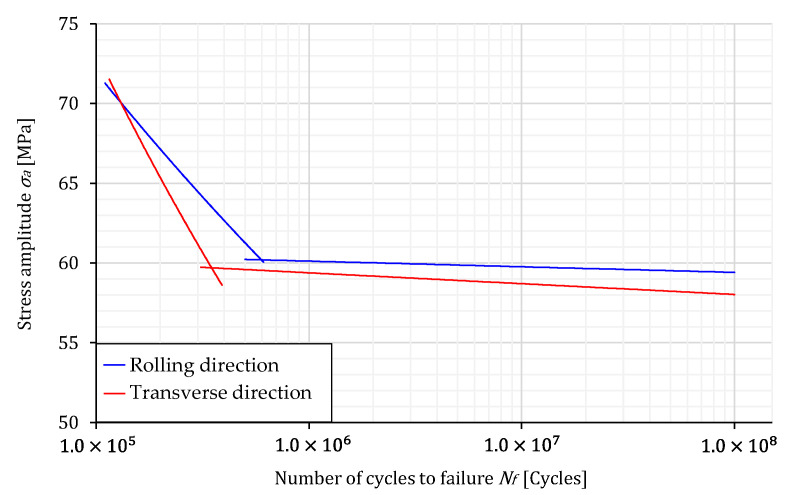
The comparison of S–N curve for both specimen’s directions.

**Figure 9 materials-16-02674-f009:**
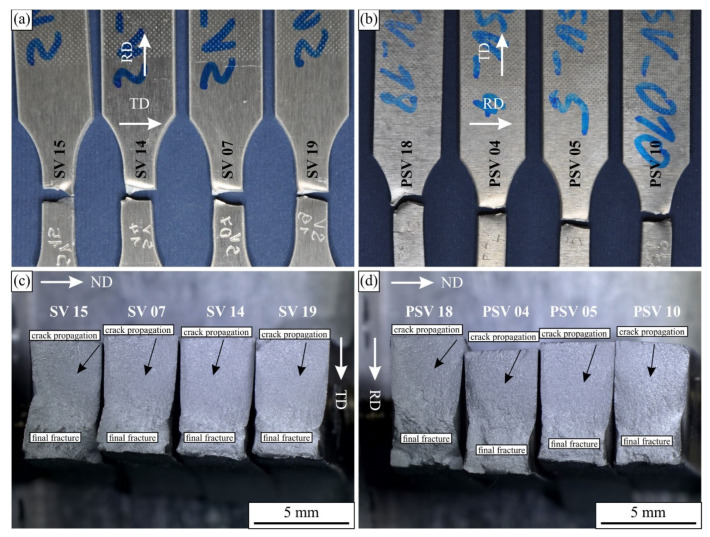
The light micrographs of the fractured surfaces of the selected fatigue samples: (**a**,**c**) longitudinal layout and (**b**,**d**) transverse layout (ND—normal direction, RD—rolling direction, TD—transverse direction).

**Figure 10 materials-16-02674-f010:**
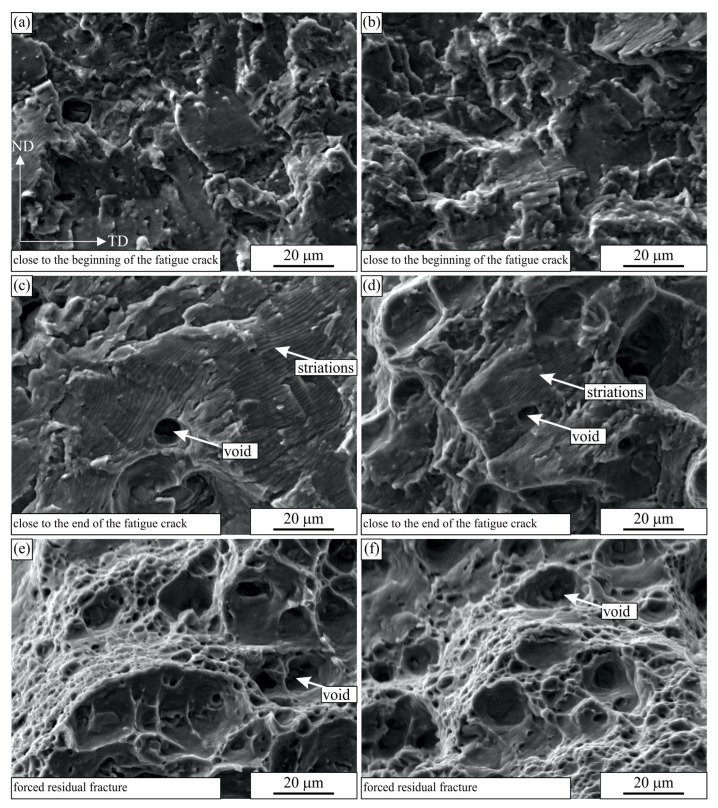
The secondary electron micrographs of the fractured surfaces of the fatigue samples tested in the longitudinal layout: (**a**,**c**,**e**) SV 15 (131,581 cycles to failure), (**b**,**d**,**f**) SV 19 (19,392,180 cycles to failure). The orientation of all images is the same. The RD direction is perpendicular to the images.

**Figure 11 materials-16-02674-f011:**
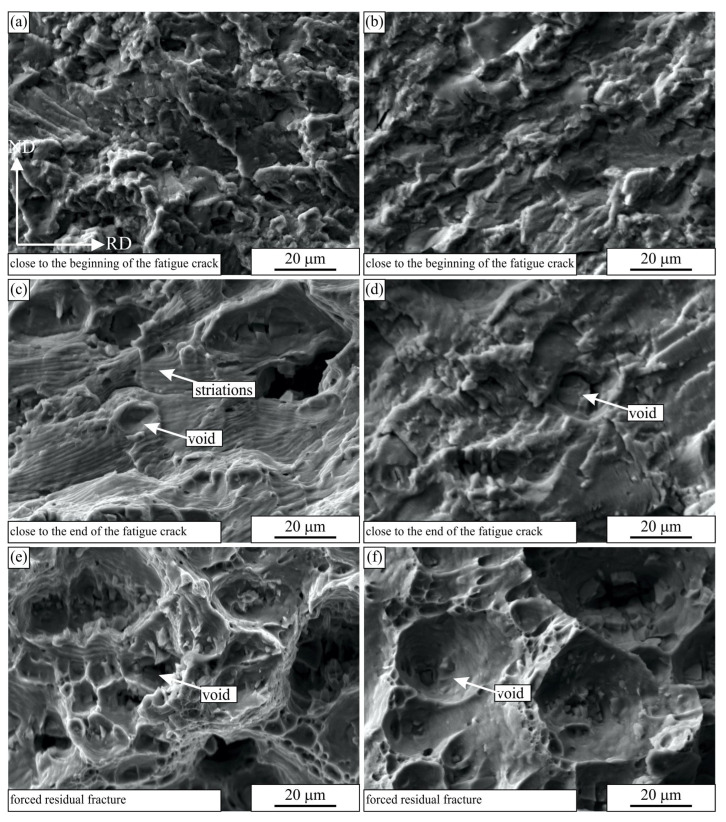
The secondary electron micrographs of the fractured surfaces of the fatigue samples tested in the transverse layout: (**a**,**c**,**e**) PSV 18 (135,165 cycles to failure), (**b**,**d**,**f**) SV 10 (15,500,000 cycles to failure). The orientation of all images is the same. The TD direction is perpendicular to the images.

**Figure 12 materials-16-02674-f012:**
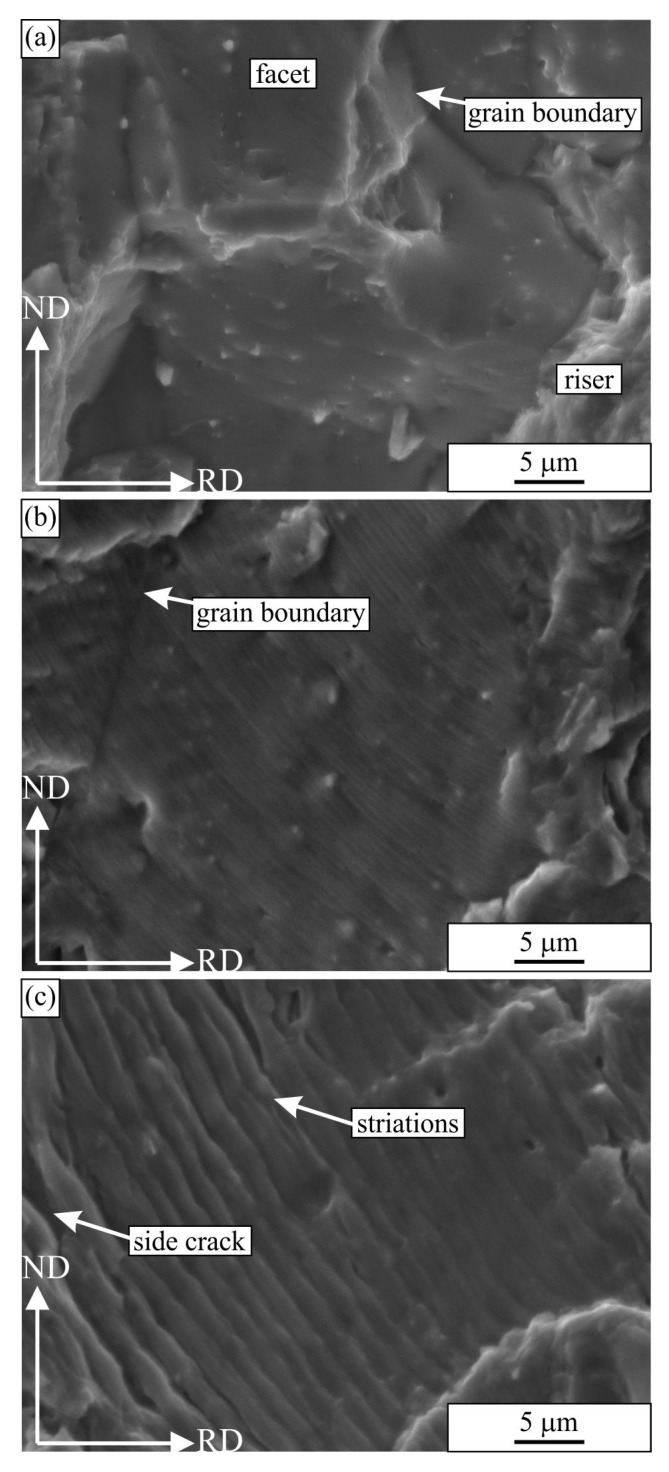
Secondary electron micrographs of the fractured surfaces of the fatigue sample PSV 14 (765,121 cycles to failure): (**a**) 0.7 mm from the fatigue crack initiation (Stage Regime A), (**b**) 1.4 mm from the fatigue crack initiation (start of Regime B), (**c**) 3.5 mm from the fatigue crack initiation (near the end of Regime B).

**Figure 13 materials-16-02674-f013:**
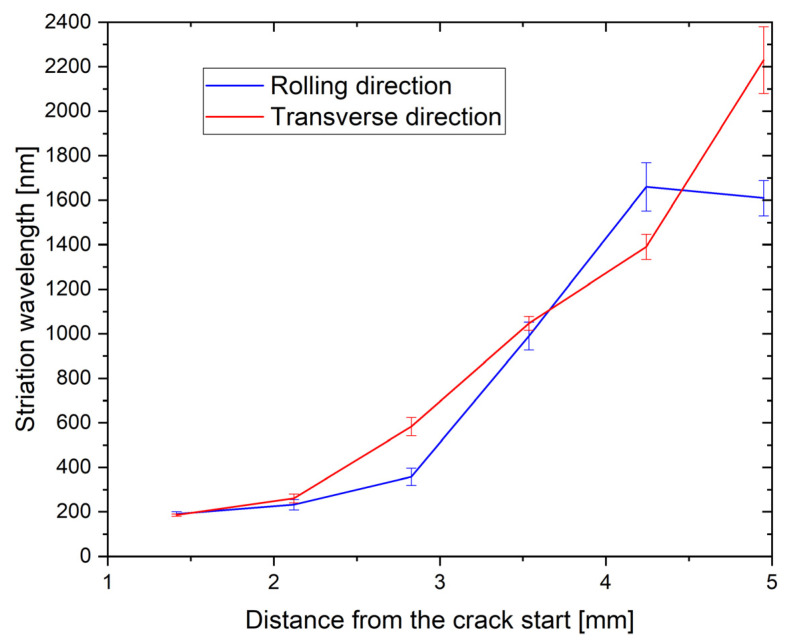
Striation wavelength in dependence from the crack start (RD—sample, TD—sample).

**Table 1 materials-16-02674-t001:** The chemical composition of AA 5083-H111.

Al [wt %]	Mg [wt %]	Mn [wt %]	Si [wt %]	Fe [wt %]	Zn [wt %]	Cr [wt %]	Ti [wt %]	Cu [wt %]
92.55	4.9	1	0.4	0.4	0.25	0.25	0.15	0.1

**Table 2 materials-16-02674-t002:** Average mechanical properties of AA 5083-H111.

Young’s Modulus *E* [GPa]		Yield Stress $\sigma_{y}$ [MPa]		UTS $\sigma_{UTS}$ [MPa]		Elongation *A* [%]	
(RD)	(TD)	(RD)	(TD)	(RD)	(TD)	(RD)	(TD)
70.8	71.2	143.2	141.9	293.9	284.2	22.2	21.9

(RD)—rolling direction, (TD)—transverse direction.

**Table 3 materials-16-02674-t003:** Fatigue parameters of AA 5083-H111 related to the bilinear S–N curve.

Number of Cycles at the Knee Point *N_k_* [Cycles]		Stress Amplitude at the Knee Point σ*_a,k_* [MPa]		Slope before the Knee Point *k* [/]		Slope after the Knee Point *K*^∗^ [/]	
**(RD)**	**(TD)**	**(RD)**	**(TD)**	**(RD)**	**(TD)**	**(RD)**	**(TD)**
596,500	348,500	60.2	59.7	10.02	6.14	389.04	199.5

(RD)—rolling direction; (TD)—transverse direction.

## Data Availability

The data presented in this study are available on request from the corresponding author.

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
