# Peer review of "High-Cycle Fatigue Behaviour of the Aluminium Alloy 5083-H111"

_materials, 2023, doi:10.3390/ma16072674_

Round 1

Reviewer 1 Report

The manuscript is about the high cycle fatigue behaviour of the aluminium alloy 5083-H111. English language and style are fine. Introduction provides sufficient background and include all relevant references. Graphical illustration of method is well designed.

I appreciate the effort that the authors have put in performing this study. I have no objections against the work by essence, but I have some comments concerning the improvement of article. It would improve the quality of the paper if the authors were willing to incorporate the following changes:

(1) It is strongly recommended authors to add visual research gap and identify the novelty of the manuscript. It can be placed after introduction section as a separate section.

(2) Please consider reviewing the abstract and highlight the novelty, major findings, and conclusions. I suggest reorganizing the abstract. Please use numbers or % terms to clearly shows us the results in your experimental work. When reading the abstract, it is all generic findings and nothing specific using numbers or % terms to clearly tell us the effects of analyzed metrics.

(3) For Table 1, it is required to indicate unit (wt%)

(4) How can authors be sure about white colored phases in Figure 2? Is it possible that some of them are oxide phases? It is recommended to add a more detailed explanation and literature information about the relevant Figure.

(5) In SEM images of microstructure or fractured surfaces, using arrows can be better to show “void”, “striations”

(6) At the first paragraph of “Conclusion” section, authors say,

 “The comprehensive experimental      …… 66 Hz up to the final failure of the specimen.”

It is repetition of abstract or methods. It is not necessary to present again.

Author Response

The manuscript is about the high cycle fatigue behaviour of the aluminium alloy 5083-H111. English language and style are fine. Introduction provides sufficient background and include all relevant references. Graphical illustration of method is well designed.

I appreciate the effort that the authors have put in performing this study. I have no objections against the work by essence, but I have some comments concerning the improvement of article. It would improve the quality of the paper if the authors were willing to incorporate the following changes:

Comment #1: It is strongly recommended authors to add visual research gap and identify the novelty of the manuscript. It can be placed after introduction section as a separate section.

Response: Experimental cyclic material testing is important for the individual rolling direction of aluminium alloy 5083-H111 because it helps to understand the material's behaviour and response under cyclic loading. In each rolling direction, the testing provides valuable information on the material's mechanical properties, such as its stress-strain response, deformation, fatigue life, and fracture behaviour. This information is crucial for optimising the design and manufacture of structures and components made from this material and assessing their performance and durability in various applications. Additionally, the results of cyclic material testing can be used to develop predictive models that can help engineers make informed decisions about the use and performance of aluminium alloy 5083-H111 in various applications. This explanation is now added at the end of the Introduction of the revised manuscript.

Comment #2: Please consider reviewing the abstract and highlight the novelty, major findings, and conclusions. I suggest reorganising the abstract. Please use numbers or % terms to clearly shows us the results in your experimental work. When reading the abstract, it is all generic findings and nothing specific using numbers or % terms to clearly tell us the effects of analysed metrics.

Response: The Abstract has been revised as suggested.

Comment #3: For Table 1, it is required to indicate unit (wt%).

Response: The unit in Table 1 was corrected as suggested.

Comment #4: How can authors be sure about white colored phases in Figure 2? Is it possible that some of them are oxide phases? It is recommended to add a more detailed explanation and literature information about the relevant Figure.

Response: Thank you for the remark. We carried out the detailed EDS analysis in our previous work (Ref. 34). The bright phases are rich in Mn and Fe, which contribute to a higher Z-contrast in the BSE images. Oxides are typically slightly darker than Al-matrix, not so much as dark Mg2Si particles. Oxides may be present in the alloy, but their fraction is very low. In this work, we carried out a few spot EDS analyses; all bright particles contained Fe and Mg, and dark particles were rich in Mg and Si. Since Mg2Si is very reactive, also some oxygen was found together with Mg and Si.

Comment #5: In SEM images of microstructure or fractured surfaces, using arrows can be better to show “void”, “striations”.

Response: Thank you for your comment. We indicated the features of interest with arrows.

Comment #6: At the first paragraph of “Conclusion” section, authors say, “The comprehensive experimental  …    …… 66 Hz up to the final failure of the specimen.” It is repetition of abstract or methods. It is not necessary to present again.”.

Response: The text was corrected (reduced) as suggested.

Reviewer 2 Report

In this manuscript, the authors reported a comprehensive experimental investigation of the High Cycle Fatigue 7(HCF) behavior of the ductile aluminum alloy AA 5083-H111. While this paper was poorly prepared without clear innovativeness in the whole paper, the content is quite solid. Therefore, the reviewer suggests minor revision the manuscript in its present state, and the following points should be considered by the authors:

1. The abstract and conclusions section should be presented brief and clear form. There is no need to list some unimportant contents, such as the preparation of samples.

2. The research considers the influence of the rolling direction on the fatigue life and fracture behavior in presented work, but they were not sufficiently presented in the conclusions.

3. Although extensive experiments were carried out, it was lacked some substantive content of the discussion part.

4. There should be a clear distinction between the abstract section and conclusions. In addition, references 13 and 35 are same, which should be carefully examined.

Author Response

1 Introduction

[Line 42-45]: The authors stated, "The tensile strength and fatigue life of Al-alloys are affected

slightly by the rolling direction at room temperature, but the effects become more significant at

higher temperatures due to the expansion of the grin structure [20, 21]. It seems like rolling

directions effect, for both room temperature and high temperature effects, on tensile strength

and fatigue behavior are already known, what is the contribution of this paper?

[Line 42-45]: Also, through that reference, the temperature is a critical factor in your experiment,

but in your fatigue testing, there is no mention of the testing temperature.

Response: A novelty of the proposed manuscript is the experimental fatigue testing of aluminium alloy 5083-H111 considering the influence of the rolling direction on the fatigue life and the detailed fractography of the cross-sections of the tested samples. The HCF fatigue tests in a load control at a load ratio of 0.1 were performed for each specimen layout. The fatigue tests were performed at a room temperature of 22°C. The novelty and the final findings of our research are now included in the Abstract and Conclusions of the revised manuscript.

2 Materials and Methods

Comment #1: [Line 113]: Did you only examine one batch material, or did you buy material from different batches to come to that Conclusion? You are making a blanket statement on microstructural observation that may not be true across all the tested alloy.

Response: The tensile specimens were taken from one batch of material. This is explained additionally in Section 2.1 of the revised manuscript. 

Comment #2: In Figure 2, how did you find each particle looked like that? Did you conduct the EDX analysis?

Response: Thank you for the remark. We carried out the detailed EDS analysis in our previous work (Ref. 34). The bright phases are rich in Mn and Fe, which contribute to a higher Z-contrast in the BSE images. Oxides are typically slightly darker than Al-matrix, not so much as dark Mg2Si particles. Oxides may be present in the alloy, but their fraction is very low. In this work, we carried out few spot EDS analyses; all bright particles contained Fe and Mg, and dark particles were rich in Mg and Si. Since Mg2Si is very reactive, also some oxygen was found together with Mg and Si.

Comment #3: [Line 133] You stopped the test when the “stiffness” dropped by 20%, how did you measure stiffness? Most testing system can only test for either displacement or load.

Response: Thank you for this comment. We mistakenly wrote that the tests were stopped when the stiffness dropped by 20 %. However, this condition can be monitored by using the extensometer, which was not the case in our study. In our experiments, at least two specimens were tested for each stress level up to the final failure of the specimen. The run-out condition (without failure) was set to 108 cycles. This explanation is now added in Section 2.2 of the revised manuscript.

Comment #4: [Line 132-133]: Only two samples were made for each load level? That is not sufficient number of samples for HCF. See also additional comment in results. How many stress levels? How are the stress levels deteremined?

Response: As already explained in the previous comment, eight different stress levels were selected for each specimen orientation. At least two specimens were tested for each stress level up to the final failure of the specimen. The run-out condition (without failure) was set to 108 cycles. The first stress level (maximum force Fmax) was defined at approximately 50 % of the yield stress of the material, and each next stress level was reduced by 3%. This explanation is added in Section 2.2 of the revised manuscript.

Comment #5: [Line 135-136]: It needs to be clarified. Is that meaning your experiments were not consistent and reliable?

Response: The scatter of experimentally determined cycles to failure in aluminium alloy samples can occur for various reasons, including:

- Material variability: Due to the presence of variations in the microstructure, properties and composition of the aluminium alloy samples, the results can show scatter.

- Loading conditions: The scatter in the results can be due to variations in loading conditions, such as temperature, loading rate, and amplitude, which can affect the material response.

- Sample preparation: The scatter in the results can also be due to variations in sample preparation, such as surface finish, thickness, and homogeneity, which can affect the material response.

- Testing methodology: The scatter in the results can also be due to variations in testing methodology, such as the use of different testing machines, testing setups, and instrumentation, which can affect measurement accuracy.

In our case, the scatter of results is probably due to the geometrical deviations between individual specimens produced by the water jet cutting technique, which is now explained in the revised version of the manuscript.

Comment #6: Have the authors considered one more orientation? For the 45-degree sample cut? If not why.

Response: In the proposed research work, we analysed only two specimen orientations: (i) in the rolling direction and (ii) transverse to the rolling direction. The third specimen orientation (45° in regard to the rolling direction) could be investigated in our further research work. This is also mentioned in the Conclusions of the revised manuscript.

3 Results and discussion

Comment #1: Table 2 and discussion related to the table: Have you done a statistical analysis of your results? The standard deviation of 4 tests per quasi-static tensile test is not expected to be a high number (evident from Figure 3), and the presented data in Table 2 does not seems congruent with authors’ statement; that is I do not think that there is a statistical difference in the RD and TD results.

Response: When analysing the experimental results by quasi-static tests, the appropriate material parameters were obtained separately for each static engineering stress-strain curve. Therefore, the average values were considered in Table 2. However, it is evident from Figure 3 that the differences between individual tests are very small.

Comment #2: [Line 173]: ‘some particles were found occasionally’. Do they have an effect on the fatigue / tensile behavior?

Response: Generally, all hard particles influence the fracture behaviour of the analysed material. Larger particles produce large stress concentrations and crack formation by decohesion at the particle/matrix interface or their fracture. In our study, large particles formed large voids, and almost in every void examined, the debris of intermetallic particles was found. On the other hand, in smaller voids, intermetallic particles were rarely found. Thus the word “occasionally” refers to the presence of smaller particles in smaller voids.

Comment #3: It is unclear what is added by this work when compared with the authors’ previous work as all the tested data points (from both studies) are lumped into Figure 5 and 6. Also, how did the authors do the curve fit? What is the R2 of the curve fit?

Response: In the author's previous paper [34], low-cycle fatigue (LCF) tests of aluminium alloy AA 5083-H111 were studied considering only one specimen orientation (in the rolling direction). In the proposed paper, high-cycle fatigue (HCF) tests of the same alloy were analysed considering two specimen orientations: (i) in the rolling direction and (ii) transverse to the rolling direction. Furthermore, in our previous work, the LCF tests were performed in strain control up to the 104 loading cycles. In the proposed paper, the HCF tests were performed in force control, and the obtained fatigue life varied between 105 and 108 loading cycles.

In Figures 5 and 6, each point represents the fatigue failure of the tested specimen. As mentioned in the manuscript, for aluminium alloys, the S-N curve consists of two parts, i.e.  "bilinear" S-N curve. The curve fit was plotted separately for each part of the S-N curve, where the trend line was plotted through all experimental points using Microsoft excel software. The R2 value of the curve fit is now added in Figures 5 and 6 of the revised manuscript.

Comment #4: Perhaps it would be appropriate to show the fractured specimen on the side vide to indicate where the failure occurred (something similar to Figure 1). Is there any significant difference in the macro fracture surface for RD vs TD (in Figure 7)? Explain?

Response: We added fractured samples, both for tensile and fatigue tests. There were no great differences in the macroscopic appearances between both directions.

Comment #5: [Line 239]: Did the author inspect the grain boundary in Figures 8 and 9? How?

Response: Fracture facets at the beginning of fatigue crack propagation had a size of approximately 20 mm, which was also the approximate size of crystal grains, found in our previous study (Ref. 34). Thus, we believe that there is a strong correlation between these values. Otherwise, it is difficult to determine crystal orientations on ductile fracture surfaces, even by the use of electron back-scattered diffraction (EBSD).

Comment #6: It would be helpful if the authors indicated the rolling directions on the SEM pictures. Also, since striations are so clearly shown on the figure, and the authors measure them, why did they not find the parameters in Paris’ law?

Response: The rolling directions are now indicated in the micrographs.

Reviewer 3 Report

Review of Materials-2217861 – High cycle fatigue behaviour of the aluminium alloy 5083-H111

General Comments

This paper presents results about the High Cycle Fatigue behavior of the ductile aluminum alloy AA 5083-H111. The S-N curves were obtained and analyzed to obtain the slope and knee point for each specimen layout. Although the topic is interesting, some sections are not explained very well. That said, the reviewer believes that this paper requires major revision.

Introduction

i.                    It is not clear what is the novelty of this work. Please explain the novelty of this work clearly at the end of introduction.

ii.                  The methodology (obtaining the S-N curves from HCF test) has been extensively used for a wide variety of materials. Please explain what is specific regarding this study.

Results and discussion

i.                    Figures 8, 9, and 10 show the fractographs of the samples. The text boxes provided in each figure (e.g., void and striations) and explanations in the text seem trivial and obvious and can be found easily in fracture textbooks and handbooks. These are the features that are expected for ductile materials. Please explain what is specific regarding this particular alloy that is different from other aluminum alloys.

Conclusions

i.                    The reviewer believes that the Conclusions section has not been written well and needs to be reworked.

ii.                  The first conclusion presented information about composition of particles in AA 5083-H111 alloy. It is not clear what is the impact of longer Fe-rich Al6(Fe, Mn) particles that are obtained from longitudinal.

iii.                The second bullet point could be expected even without conducting the experiments because of the cold working during the rolling process. Please explain if a different result was expected for this alloy.

iv.                The fourth bullet point could also be expected without conducting the experiment as the material properties that were obtained for both layouts were close to one another. The elongation of grains during the rolling process was not analyzed in this study. The impact of the rolling process on the material properties can be understood better if the extent of elongation of grains is known.

Author Response

The submitted manuscript for review concerns the presentation of the results of fatigue tests on ductile aluminum alloy AA5083-H111. In the paper, the authors presented the results of HCF tests of this material based on two sets of samples taken in line with and in order with respect to the rolling direction of the material (sheet).

The work is free of significant errors, however, its scientific soundness is low - after the proposed changes, the work will be able to be recommended for publication. Comments on the submitted manuscript:

Comment #1: There is a lack of emphasis on the element of scientific novelty in the manuscript, it was difficult for the reviewer to bring it out - this should be corrected?

Response: The following paragraph has been added at the end of the Introduction:

Material testing provides valuable information on the material's mechanical properties, such as its stress-strain response, deformation, fatigue life, and fracture behaviour. This information is critical for optimising the design and manufacture of engineering structures and components made from this material and assessing their performance and durability in various applications. The proposed study is the continuation of the author's previous work [34], where the LCF behaviour of the aluminium alloy 5083-H111 was investigated. The presented work focuses on the HCF behaviour of the same alloy. However, the proposed research also considers the influence of the rolling direction on the fatigue life and fracture behaviour of analysed Al-alloy in the high cycle fatigue regime, considering the comprehensive fractography of both quasi-static and fatigue specimens. Additionally, the obtained results could help engineers make the appropriate decisions about the use and performance of AA 5083-H111 in various engineering applications.

Comment #2: In the manuscript, the authors draw attention to the variability of the striations observed in the SEM images, which was not referenced in the conclusions.

Response: The following paragraph has been added in the Conclusion of the revised manuscript:

“The fracture surface has a typical appearance for the ductile material, characterised by striations during propagation of the fatigue crack and final ductile fracture. The distance between striations increased from the crack beginning (less than 0.5 micrometres) to the crack end (more than 3 micrometres)”.

Comment #3: I suggest paying special attention to the fractographic part of the analysis carried out - here the direction of the orientation of the specimens relative to the rolling direction can be visualised in fatigue fractures?

Response:  On all fractographic images, the normal direction (ND) was always vertical. The horizontal direction was either the rolling direction (RD) or transverse direction (TD), which is now indicated on the corresponding images in the revised manuscript.

Reviewer 4 Report

Please see comment in PDF.

Author Response

This paper presents results about the High Cycle Fatigue behavior of the ductile aluminum alloy AA 5083-H111. The S-N curves were obtained and analysed to obtain the slope and knee point for each specimen layout. Although the topic is interesting, some sections are not explained very well. That said, the reviewer believes that this paper requires major revision.

1 Introduction

Comment #1: It is not clear what is the novelty of this work. Please explain the novelty of this work clearly at the end of Introduction.

Response: The following paragraph has been added at the end of the Introduction:

Material testing provides valuable information on the material's mechanical properties, such as its stress-strain response, deformation, fatigue life, and fracture behaviour. This information is critical for optimising the design and manufacture of engineering structures and components made from this material and assessing their performance and durability in various applications. The proposed study is the continuation of the author's previous work [34], where the LCF behaviour of the aluminium alloy 5083-H111 was investigated. The presented work focuses on the HCF behaviour of the same alloy. However, the proposed research also considers the influence of the rolling direction on the fatigue life and fracture behaviour of analysed Al-alloy in the high cycle fatigue regime, considering the comprehensive fractography of both quasi-static and fatigue specimens. Additionally, the obtained results could help engineers make the appropriate decisions about the use and performance of AA 5083-H111 in various engineering applications.

Comment #2: The methodology (obtaining the S-N curves from HCF test) has been extensively used for a wide variety of materials. Please explain what is specific regarding this study.

Response: As already explained in Comment 1, the proposed research considers the influence of the rolling direction on the fatigue life and fracture behaviour of analysed Al-alloy in the high cycle fatigue regime, considering the comprehensive fractography of both quasi-static and fatigue specimens.

2 Results and discussion

Comment #1: Figures 8, 9, and 10 show the fractographs of the samples. The text boxes provided in each figure (e.g., void and striations) and explanations in the text seem trivial and obvious and can be found easily in fracture textbooks and handbooks. These are the features that are expected for ductile materials. Please explain what is specific regarding this particular alloy that is different from other aluminum alloys.

Response: The main point of the investigation was to determine the fatigue behaviour of the investigated alloy in the rolling and transverse directions. The fracture characteristics of the alloy were indeed typical for the ductile fracture of metallic alloys.

3 Conclusions

Comment #1: The reviewer believes that the Conclusions section has not been written well and needs to be reworked.

Response: The Section Conclusions has been rewritten as suggested.

Comment #2: The first Conclusion presented information about composition of particles in AA 5083-H111 alloy. It is not clear what is the impact of longer Fe-rich Al6(Fe, Mn) particles that are obtained from longitudinal.

Response: In the three dimensions, the main part of the larger particles has a cuboidal shape, with a larger axis approximately parallel to the rolling direction. Thus, in the RD orientation, the larger axis of particles lay in the direction of the load and in the TD orientation perpendicular to the load. It could be expected that at TD orientation, higher stress concentrations occurred at the particle-matrix interface and that this leads to slightly worse fatigue resistance in the TD direction.

Comment #3: The second bullet point could be expected even without conducting the experiments because of the cold working during the rolling process. Please explain if a different result was expected for this alloy.

Response: The second bullet describes the obtained mechanical properties for both longitudinal/rolling (RD) and transversal (TD) specimen directions. As expected, the RD-specimens demonstrated slightly better mechanical properties if compared to the TD-specimens. The difference can be caused by large, in the rolling direction elongated Al6(Mn,Fe) particles, which cause higher stress concentrations when tested in the TD. However, this difference is more significant in the case of fatigue loading.

Comment #4: The fourth bullet point could also be expected without conducting the experiment as the material properties that were obtained for both layouts were close to one another. The elongation of grains during the rolling process was not analysed in this study. The impact of the rolling process on the material properties can be understood better if the extent of elongation of grains is known.

Response: The conclusions regarding fractured surfaces have been extended in the revised manuscript.

Reviewer 5 Report

The submitted manuscript for review concerns the presentation of the results of fatigue tests on ductile aluminum alloy AA5083-H111. In the paper, the authors presented the results of HCF tests of this material based on two sets of samples taken in line with and in order with respect to the rolling direction of the material (sheet).

Comments on the submitted manuscript:

- there is a lack of emphasis on the element of scientific novelty in the manuscript, it was difficult for the reviewer to bring it out - this should be corrected.

- in the manuscript, the authors draw attention to the variability of the striations observed in the SEM images, which was not referenced in the conclusions

- I suggest paying special attention to the fractographic part of the analysis carried out - here the direction of the orientation of the specimens relative to the rolling direction can be visualized in fatigue fractures

The work is free of significant errors, however, its scientific soundness is low - after the proposed changes, the work will be able to be recommended for publication.

Author Response

In this manuscript, the authors reported a comprehensive experimental investigation of the High Cycle Fatigue 7(HCF) behavior of the ductile aluminum alloy AA 5083-H111. While this paper was poorly prepared without clear innovativeness in the whole paper, the content is quite solid. Therefore, the reviewer suggests minor revision the manuscript in its present state, and the following points should be considered by the authors.

Comment #1: The abstract and conclusions section should be presented brief and clear form. There is no need to list some unimportant contents, such as the preparation of samples.

Response: The Abstract has been rewritten as suggested.

Comment #2: The research considers the influence of the rolling direction on the fatigue life and fracture behavior in presented work, but they were not sufficiently presented in the conclusions.

Response: The Conclusion has been rewritten as suggested.

Comment #3: Although extensive experiments were carried out, it was lacked some substantive content of the discussion part.

Response: Section 3 (Results and discussion) has been rewritten as suggested.

Comment #4: There should be a clear distinction between the abstract section and conclusions. In addition, references 13 and 35 are same, which should be carefully examined.

Response: Both the Abstract and Conclusion sections have been rewritten. Reference [35] has been omitted.

Round 2

Reviewer 3 Report

Thank you for providing detailed responses to some of the questions. The reviewer still believes that the element of novelty, which is the most important element of any research, is missing in this work. The methodology used in this work (obtaining S-N curves for HCF) has been studied extensively by many researchers. Changing the grade of material while similar behavior and fracture surfaces are expected does not bring about novel results. Although it was asked by the reviewer in the first round of review, the authors have not explained why this specific material and grade has been used for this study and what makes this work different from the other researchers already conducted. Are any of the results obtained from this material significantly different from those of other aluminum alloys? Please explain.

Author Response

Comment: Thank you for providing detailed responses to some of the questions. The reviewer still believes that the element of novelty, which is the most important element of any research, is missing in this work. The methodology used in this work (obtaining S-N curves for HCF) has been studied extensively by many researchers. Changing the grade of material while similar behavior and fracture surfaces are expected does not bring about novel results. Although it was asked by the reviewer in the first round of review, the authors have not explained why this specific material and grade has been used for this study and what makes this work different from the other researchers already conducted. Are any of the results obtained from this material significantly different from those of other aluminum alloys? Please explain.

Response: As explained additionally in Revised version 2 of the manuscript (see last paragraph in Introduction), the proposed study is the continuation of our previous work [34], where the LCF behaviour of the aluminium alloy 5083-H111 was investigated. The obtained experimental results (cyclic S-N curve, LCF-fatigue parameters) were then used to study the LCF behaviour of cellular structures (we added a new Fig. 1 in the revised manuscript), which represent a unique opportunity for adoption in lightweight design due to their favourable characteristics regarding sound isolation, damping, energy absorption, etc. When analysing the fatigue behaviour of cellular structures made of aluminium alloy AA 5083-H111, the effect of rolling direction on the fatigue life may be significant, especially in the High Cycle Fatigue (HCF) area. For that reason, we decided to investigate the influence of the rolling direction on the fatigue life and fracture behaviour of aluminium alloy AA 5083-H111 in the HCF regime. The obtained results will help us in subsequent computational fatigue analyses of cellular structures in the HCF area. This explanation and, consequently, the motivation for our research is now pointed out in the last paragraph in the section Introduction of the revised manuscript.

Reviewer 4 Report

The authors did not address any of my concerns on

1) distinction between previous paper data (LCF) and data from this paper HCF

2) Too few data points are used to come to a conclusion (8 tests?) for HCF

3) did not make an attempt to quantify the striations and how its influence by rolling direction (or not influenced by rolling direction).

4) Curve fitting parameters (R^2) are not shown to indicate the reliability of the observations.

5) You need a lot more than 1 runout.  Load levels/procedures to determine runout are not clearly documented

Author Response

Comment #1: Distinction between previous paper data (LCF) and data from this paper HCF.

Response: In the author's previous paper [34], low-cycle fatigue (LCF) tests of aluminium alloy AA 5083-H111 were studied considering only one specimen orientation (in the rolling direction). In the proposed paper, high-cycle fatigue (HCF) tests of the same alloy were analysed considering two specimen orientations: (i) in the rolling direction and (ii) transverse to the rolling direction. Furthermore, in our previous work, the LCF tests were performed in strain control up to the 104 loading cycles. In the proposed paper, the HCF tests were performed in force control, and the obtained fatigue life varied between 105 and 108 loading cycles. This information is now included in the revised version of the manuscript.

Comment #2: Too few data points are used to come to a conclusion (8 tests?) for HCF.

Response: In the scope of experimental fatigue investigation, 15 HCF tests were performed to determine the S-N curve in the rolling direction, and 20 HCF tests transversal to the rolling direction. As already explained in the manuscript,  eight different stress levels were selected, and at least two specimens were tested at each stress level up to the final failure of the specimen.

Comment #3: Did not make an attempt to quantify the striations and how its influence by rolling direction (or not influenced by rolling direction).

Response: We have done this measurement additionally, and the results are presented in Figure 12 in the revised manuscript. It was found that during the main part of the fatigue crack, the striations have larger wavelengths in the transverse direction.

Comment #4: Curve fitting parameters (R^2) are not shown to indicate the reliability of the observations.

Response: In the revised manuscript, Figures 6 and 7 have been corrected by adding the fitting parameters of S-N curves.

Comment #5: You need a lot more than 1 runout. Load levels/procedures to determine runout are not clearly documented.

Response: The authors agree with the reviewer that more than one runout test should be done to describe the fatigue behaviour in this area more exactly. However, we will need about three weeks for one such test. For that reason, this could be the plan for our further investigations.  

Reviewer 5 Report

The manuscript is now properly prepared, I accept the changes made, which raised the quality of the manuscript.  The article is suitable for publication in Materials journal

Author Response

Comment: The manuscript is now properly prepared, I accept the changes made, which raised the quality of the manuscript.  The article is suitable for publication in Materials journal.

Response: Thank you very much.

Round 3

Reviewer 3 Report

Thanks for your response. 

Author Response

Thank you.

Reviewer 4 Report

It is still not ready for publication.  Please see attached comment

Author Response

Comment #1: Line 179-184: Authors claimed that mechanical properties are better in the RD direction than TD, yet the elastic modulus in the TD direction is higher (Table 2), explain.

Response: Thank you for this comment. The authors agree with the Reviewer that the description of the experimental results of quasi-static tests was not precise enough in Revision-2 of our manuscript. It is clear that there were no significant differences in mechanical properties for both RD and TD specimens. For that reason, we changed this paragraph as follows:

Based on the experimental results presented in Figure 4 and Table 2, it can be concluded that the mechanical properties (Young’s modulus, yield stress, ultimate tensile strength, elongation at fracture) were quite similar for both specimens. However, the specimens manufactured in the longitudinal/rolling direction demonstrated slightly better properties (except for Young’s modulus E) if compared to the specimens manufactured perpendicular to the rolling direction (transverse direction).

Comment #2: Again, I mentioned earlier that test data standard deviation (and ANOVA analysis) needs to be reported to show that there is a difference in the RD and TD results.

Response: As explained in Comments 3 and 4, some additional experiments have been done for both specimens' layouts. However, these experiments were related to higher stresses and, consequently, to shorter fatigue lives (we would need a few months to finish additional experiments in the long-life fatigue area). For that reason, we explained additionally in the Conclusion of the revised manuscript that further research work should consider the higher number of experiments, especially in the long-life fatigue area (more than 107 loading cycles). In this case, a comprehensive statistical evaluation could be performed to get more qualitative results regarding the fatigue behaviour of the analysed aluminium alloy. Besides this explanation, we also added a new Figure in Section 3.2 (Figure 8 in the revised manuscript), where S-N curves for both specimens' layouts are presented in the same diagram. It is evident from Figure 8 that RD-specimens demonstrated longer fatigue life if compared to TD-specimens.

Comment #3: Figure 5: there are only 6 data points plotted for the HCF part of this curve, yet the authors claimed a lot more HCF tests were done, please explain. Again, 1 run out is NOT sufficient for HCF testing.

Response: The authors agree with the Reviewer that more experiments should be done to get more representative results. For that reason, a few additional experiments were performed under high stresses where the test duration was relatively short. Eight additional tests were performed for the RD specimen layout, and two tests were performed for the TD specimen layout. The obtained experimental results are summarised in the table below. In the revised manuscript, the obtained experimental data points were included in Figures 6 and 7.

Rolling direction

Amplitude stress

Number of cycles to failure

Test duration

71.2

105397

0.44 h

70.7

115839

0.49 h

68.3

150305

0.63 h

64.5

285008

1.20 h

63.0

415004

1.75 h

63.0

391897

1.65 h

60.0

3702486

15.58 h

59.8

11101457

46.72 h

Transverse direction

Amplitude stress

Number of cycles to failure

Test duration

60.7

350158

1.47 h

59.1

1845135

7.77 h

The authors agree with the Reviewer that only one run-out test is not enough. As defined in the manuscript, the run-out test was set to 108 cycles. Considering the applied loading frequency of 66 Hz, the duration of each run-out test will be 17.5 days which means that we will need two or three months to perform some additional run-out tests. For that reason, the following explanation has been added to the Conclusion of the revised manuscript:

In the proposed research work, we analysed only two specimen orientations: (i) in the rolling direction and (ii) transverse to the rolling direction. The third specimen orientation (45° in regard to the rolling direction) could be investigated in our further research work. Furthermore, further research work should consider the higher number of experiments, especially in the long-life fatigue area (more than 107 loading cycles). In this case, a comprehensive statistical evaluation could be performed to get more qualitative results regarding the fatigue behaviour of the analysed aluminium alloy.

Comment #4: Figures 5 and 6: R^2 for the HCF is about 0.5, that is not a good curve fit.

Response: Considering the additional experiment presented in the table above, the curve fit in the HCF-area was slightly improved (from 0.56 to 0.567 for the RD specimens and from 0.5 to 0.548 for the TD specimens). As already explained in Comment #3, some additional tests should be performed in further research work to get more qualitative results regarding the fatigue behaviour of the analysed aluminium alloy. 

Comment #5: Figure 5: at load level of about 59.5 MPa, two data points were shown: one failed at 5 million cycles, the other run out at 100 million cycles. Clearly, more testing is needed. In line 147, the authors claimed “A third specimen was tested if the difference between the cycles to failure of the tested specimens was greater than 40 % of their average values,” yet there are only two data points, and it is nowhere near 40% of the average value that the authors are claiming.The same is true for Figure 6.

Response: As already explained in Comment #3, eight additional tests were performed for the RD specimen layout, and two tests were performed for the TD specimen layout. The obtained experimental results were then included in Figures 6 and 7, where the S-N curves of both specimen directions are presented. The authors agree with the Reviewer that the diction “A third specimen was tested if the difference between the cycles to failure of the tested specimens was greater than 40 % of their average values” is probably unclear. For that reason, we changed this diction in the revised manuscript as follows: At least two or three specimens were tested for each stress level up to the final failure of the specimen.

Comment #6: In describing striations at the beginning region to the end region of cracked surface, the authors did not discuss why there is a difference between the two regions.

Response: We have replaced the high-resolution images in Figure 11 (Figure 12 in the revised manuscript) with new ones, which better represent the appearance of the fracture surfaces at different stages of fatigue crack propagation. We have also provided the following discussion when referring to Figures 12 and 13 in the revised manuscript:

Figure 12 shows some fractured surfaces at a much higher magnification and resolution. The distances between the striations – the striation wavelengths - were measured on the basis of such micrographs. They are presented in Figure 13. The striations were not visible up to distances below 1 mm from the crack initiation site (Figure 12 a), which was typically at the specimen corner. This is Region A of crack propagation, which is highly sensitive to microstructure characteristics [39]. Usually, a crack with a length 1‒2 mm is considered as a small fatigue crack. One can observe grain boundaries and facets in the crystal grains, which typically glide planes. The crack propagation rate is often in the order of nm, and the specimen survives in this region for most of its lifetime [40]. Striations become visible in range B of crack propagation [39]. The striations become stable, and in the first part, their wavelength is in the order of 100 nm; we measured 200 nm at a distance of 1.4 mm from the crack initiation site (Figure 12b).With growing fatigue crack, the stress intensity factor grows, and the striation wavelength increases in micrometre size. It should be stressed that with the wavelength of 1 m, the fatigue crack will proceed 1 mm after 1000 cycles and leads to a quick fracture of the specimen. This wavelength was achieved approximately 3.5 mm from the crack initiation (Figure 12c). In Regime B, the crack propagation is less sensitive to microstructure. During the main part of the crack path, the distances between the striations were larger for the transverse specimen, which can also explain the shorter lifetime for this specimen. The last two measurements were close to the crack ends, where the surface resembled those to that shown in Figures 10c,d and Figure 11c,d, where the scatter was rather large. At the transition from Regime B to Regime C, the crack propagation becomes more microstructure sensitive. In the case of this alloy, fatigue cracks can form even in the front of the main cracks around the largest inclusions, as was discussed before.